# Impact of Explicitly Parameterized Mid-to-Low Level Latent Heating on the Simulation of a Squall Line in South China

Hai Chu [1], Mengjuan Liu [2], Leiming Ma [1,*], Xuwei Bao [2], Lanjun Zou [1,*] and Jiakai Zhu [3]

1   Shanghai Central Meteorological Observatory, Shanghai 200030, China
2   Shanghai Typhoon Institute, Shanghai 200030, China
3   Shanghai Meteorological Information and Technological Support Center, Shanghai 200030, China
*   Correspondence: malm@typhoon.org.cn (L.M.); lanjunzou@126.com (L.Z.)

**Abstract:** Organized mesoscale convective systems (MCSs), such as squall lines, are often poorly forecasted in numerical weather prediction models. In this study, experiments are performed to show that the vertical distribution of latent heating (LH) plays an important role in organizing a trailing-stratiform (TS) squall line over South China. We investigated the impact of modifying the altitude of LH peaking around 2–5 km on the squall line. It is found that increasing LH peaking at a lower vertical level (around 2–3 km) is crucial for the simulation of the TS squall line by influencing the evolution of the front-to-rear tilted upward flow and its associated mesoscale rear-to-front flow below. The influence of different LH profiles on the structure of the simulated squall line is explained using the Rotunno–Klemp–Weisman (RKW) theory considering the effects of different heights of the vertical wind center. Stronger LH at lower heights results in a vertical wind core centered lower in the convection region. Behind the core, at the mid-to-low level, is a region of descending negative horizontal vorticity. Such negative vorticity region favors a descending flow below it. When this mesoscale flow with low equivalent potential temperature ($\theta_e$) descends and catches up with the convection at near-surface, it enhances both the strength and moving speed of the convection system. Results of this study highlight the sensitivities of the MCS structure to the vertical distribution of the thermodynamical field besides traditional cold pool aspects and provide insights for the study of squall line through shear convection interaction.

**Keywords:** latent heating; squall line simulation; rear-to-front flow; shear convection interaction

## 1. Introduction

Mesoscale convective systems (MCSs) are organized storms that account for a large proportion of precipitation in tropics and warmer midlatitudes [1], thus they are important for hydrology and water supplement in vegetation. While successful numerical forecasts of organized MCS, such as squall lines, are still quite rare due to various reasons, such as inherent low predictability, deficient initial conditions, and inadequate parameterization schemes [2], studies have shown that latent heating (LH) is one important element in MCS affecting both system propagation [3,4] and organizing system circulation [5,6]. However, presently there is no direct measurement of LH profiles, and it is difficult to obtain accurate LH distribution within a mesoscale system, especially in the vertical direction. Usually, the LH profile is estimated in two ways: one is through budget calculation from a radiosonde network [7–9]; and the other is through retrieval schemes using satellite remote sensing [10–12]. The radiosonde method is usually applied in regional analyses for the limited space and time resolution of the sounding network. Alternatively, most retrieval schemes using satellite observations rely on some type of cloud-resolving models (CRMs). Overall, these retrieval schemes did well in producing the horizontal distributions of LH, but they varied in the derived altitudes of maximum heating [10,13,14], and they were more often used in long-range and large-scale studies. When it comes to detailed analysis

of the LH profile within one specific MCS case, it usually needs to turn to the help of CRM simulation in case it can reproduce the MCS event reasonably [15].

Although microphysics scheme is the main producer of LH in mesoscale simulations under 10 km resolution, uncertainties of LH may also come from other factors coupling with microphysics, such as surface sensible and latent heat fluxes related to land-surface processes, atmosphere boundary process as well as model dynamics within the convection system. Some studies have been conducted to investigate the uncertainty of convection simulations by using perturbed temperature tendency term related to LH [16–19]. Traditionally, the perturbation is spatially and temporally random [20]. However, uncertainties of the numerical model may be system-related and have spatial structure [21]. Palmer et al. [16] used a univariate Gaussian-form perturbation to obtain a space-related error for a certain horizontal scale. Qiao et al. [19] introduced stochastic perturbations to temperature tendencies to incorporate errors associated with microphysics schemes in supercell simulations. Among the ensemble experiments of Qiao et al. [19], it was found that the members with positive temperature perturbations better forecasted the convection intensity by intensifying mid-level heating near the strong updraft region. However, in designing these perturbations they were often considered only with certain horizontal scales and were not related to the convective systems. Previously, most numerical simulation researches on the effect of different LH altitudes were about large-scale circulations [22,23]. The fact that CRMs tend to create top-heavier reflectivity and overestimate MCS updrafts in the middle and upper troposphere compared to the observation in many cases [24–26], indicate that uncertainties of certain vertical scale may exist in these simulations. In their ideal simulation study, Pandya and Durran [5] indicated that differences in LH profiles may result in different development of the circulation within MCSs. However, there were a few studies about the effect of vertical distribution of LH on MCSs in real cases. In this study, a cloud-resolving model is used to study a squall line case that occurred during the first rainy season in South China. Sensitivity experiments are performed by applying modifications with certain vertical scales using the temperature tendency term from the microphysical LH to study their impact on the convection development of the squall line, especially for the relation to the rear-to-front flow, upward motion, and TS (trailing-stratiform) or LS (leading-stratiform) structure of the system.

This paper is arranged as follows: Section 2 briefly describes the squall line case and its synoptic background. Section 3 presents the model setup and the design of the sensitivity experiments. Section 4 compares the simulation results with observations and describes their different LHs. Section 5 analyzes the different convection structures, uses the horizontal vorticity method to explain them, and presents a modified conceptual model. Section 6 states the summary and discussion.

## 2. Overview of the 18–19 April 2019 Squall Line Case

The first rainy season in South China usually starts in April and lasts until June. This is the main rainy season in South China and brings about 40–50% of total annual precipitation [27,28]. From the night of April 18 to the morning of April 19 in 2019, mesoscale convection occurred continuously over the Guangdong Province and evolved into a squall line with heavy precipitation and strong gusts of wind. The squall line was present in a favorable synoptic environment for convection, with west-to-southwesterly flow at levels above 850 hPa and an underlying warm front. As shown in Figure 1, a subtropical high was maintained over the South China Sea from 18:00 UTC on April 18 through 00:00 UTC on April 19. The 500 hPa wind flow over Guangdong mostly consisted of westerlies. A west/southwesterly jet with speed over 12 ms$^{-1}$ occurred at 700 hPa and 850 hPa on April 18–19 (not shown), transporting abundant water vapor to Guangdong. Meanwhile, a west–easterly oriented warm front (dashed line) occurred at 925 hPa over Guangdong. A CAPE gradient was present across the warm front, indicating strong instability south of it.

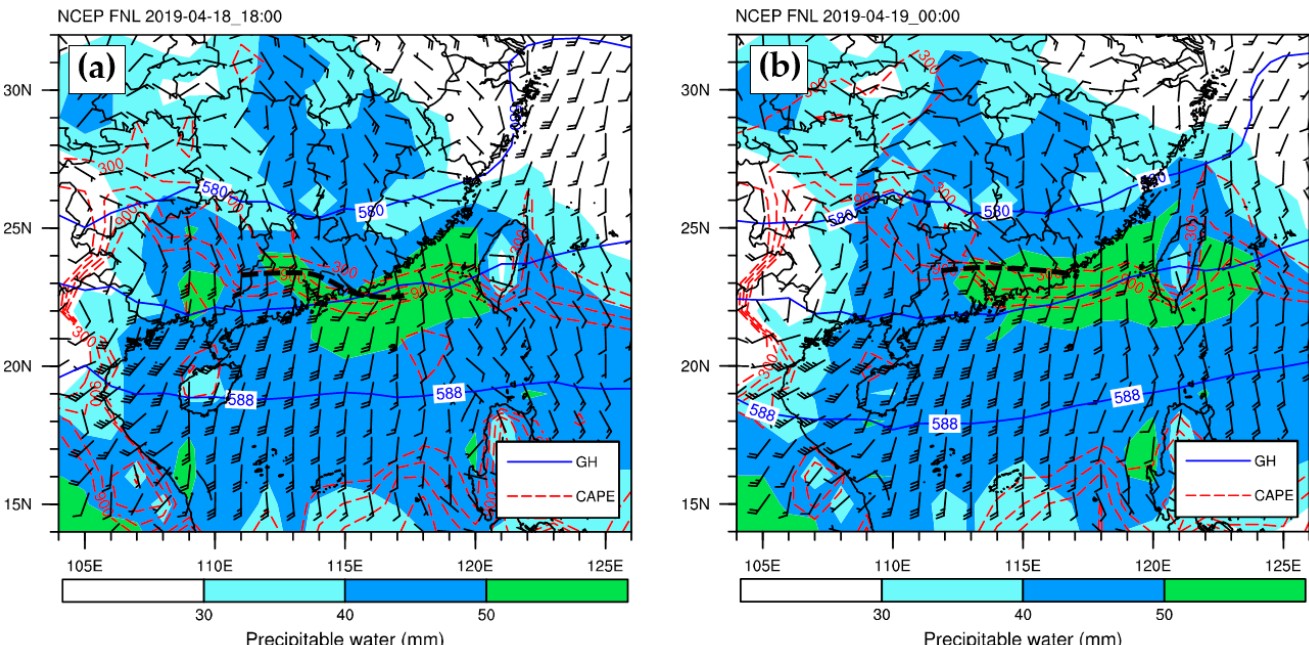

**Figure 1.** Synoptic features obtained from the National Center for Environmental Prediction Final (FNL) Operational Global Analysis data: the 500 hPa geopotential height (blue contours at 4 dagpm intervals), convective available potential energy (CAPE, red dashed line contours starting from 300 J/kg at 300 J/kg intervals), precipitable water (shaded, unit: mm) and 925 hPa wind (wind barb, a full barb is 4 ms$^{-1}$, and the warm front at this level is subjectively indicated by black dashed lines). (**a**) 18:00 UTC on April 18, (**b**) 00:00 UTC on April 19.

The squall line was oriented southwest–northeasterly, with different features identified in the western and eastern portions, demonstrated by the observations obtained at Huizhou (west) and Zijin (east) (Figure 2). Heavy hourly precipitation (24.8 mm) and strong wind gusts (16.7 ms$^{-1}$), as well as a temperature drop of 1.8 °C and a pressure rise of 1.8 hPa, were observed as the western part of the squall line passed Huizhou at approximately 22:00 UTC on April 18 (Figure 2a). The squall line also produced heavy precipitation (46.4 mm) as its eastern side passed Zijin at approximately 23:00 UTC, but wind gusts (8.3 ms$^{-1}$), temperature drop (0.3 °C) and pressure rise (0.9 hPa) measured under the eastern part were all weaker than those measured under the western part (Figure 2b).

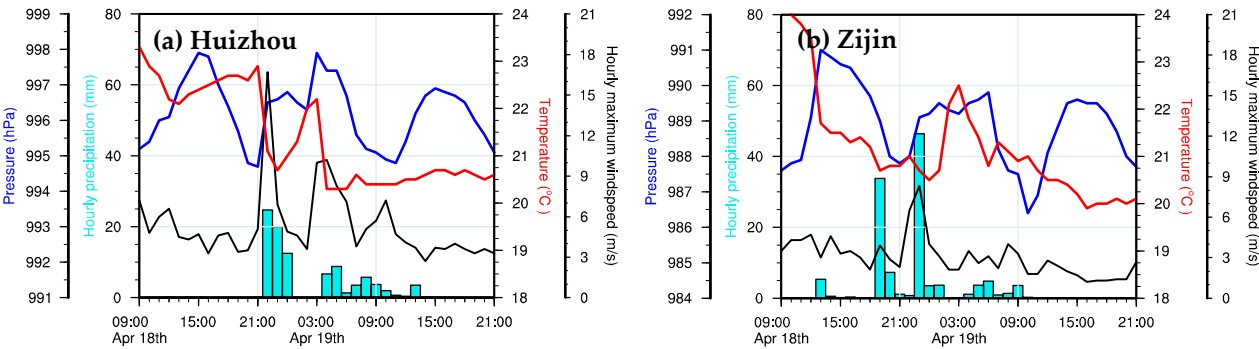

**Figure 2.** Hourly observed pressure (blue lines), precipitation (green bars), 2 m temperature (red lines) and maximum 10 m wind speed (black lines) values from 09:00 UTC on April 18 to 21:00 UTC on April 19 at (**a**) Huizhou and (**b**) Zijin.

## 3. Model Configuration and Experimental Design

### 3.1. Model Configuration

The Weather Research and Forecasting (WRF) Model [29] developed by the National Center for Atmospheric Research (NCAR) is chosen for the numerical simulations in this study. The WRF model is known as arguably the world's most widely used numerical weather prediction model both for research and real-time forecasts [30]. It is a community model with contributions of users all over the world and provides various applications in several fields, such as chemistry, hydrology, wildland fires, and regional climate. The ARW-WRF (version 3.9) model is used in this study. The horizontal model grid is $600 \times 600$ with a 3 km resolution (Figure 3). For the vertical direction, there are 40 eta mass levels in the default model setting. The National Centers for Environmental Prediction (NCEP) Final Operational Global Analysis (FNL) data, with a 1° resolution, are used as the initial and boundary conditions at 6 h intervals. A radar reflectivity assimilation is applied at the initial time (18:00 UTC on April 18) in the model using the cloud analysis method within the Community Gridpoint Statistical Interpolation system (GSI) [31] to better initiate convection. A 12 h forecast is performed to simulate the squall line from 18:00 UTC on April 18 to 06:00 UTC on April 19. The Thompson scheme [32] is used for the microphysics scheme. The unified Noah land-surface model [33,34] and the Revised MM5 Monin–Obukhov surface layer scheme [35] are used to describe near-surface model physics. Other model configurations include the scale-aware Grell–Freitas cumulus parameterization scheme [36], the UW boundary layer scheme [37], RRTM longwave radiation scheme [38], and the Duhia shortwave radiation scheme [39].

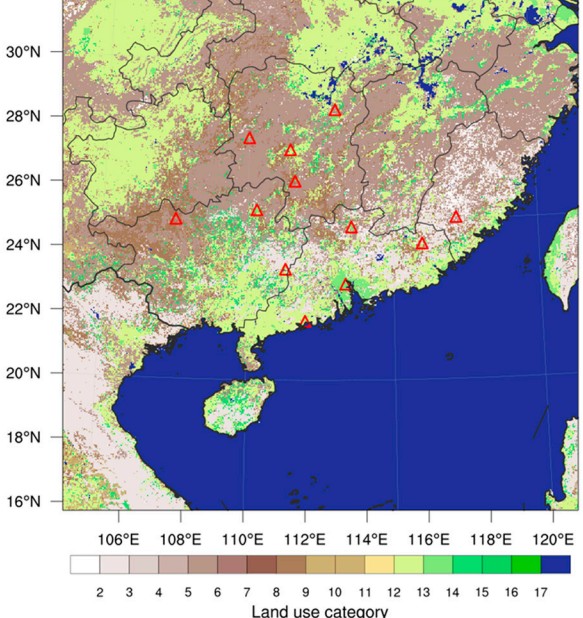

| category | Description |
| --- | --- |
| 1 | Evergreen needleleaf forest |
| 2 | Evergreen broadleaf forest |
| 3 | Deciduous needleleaf forest |
| 4 | Deciduous broadleaf forest |
| 5 | Mixed forests |
| 6 | Closed shrubland |
| 7 | open shrubland |
| 8 | Woody savannas |
| 9 | Savannas |
| 10 | Grasslands |
| 11 | Permanent Wetlands |
| 12 | Croplands |
| 13 | Urban and built-up |
| 14 | Cropland/Natural vegetation mosaic |
| 15 | Snow and ice |
| 16 | Barren or sparsely vegetated |
| 17 | Water |

**Figure 3.** Model domain with land use category. The red triangles denote the locations of radars used in the GSI cloud analysis procedure.

### 3.2. Experiment Design

In order to study the impact of different LH profiles on the convection system, we carry out sensitivity experiments with modifications in the model's thermodynamic field. This modification is applied to the temperature tendency term obtained from microphysics latent heating (*mpten*) after the microphysics scheme is called at each time step integration. Similar to the methods used by Palmer et al. [16] and Qiao et al. [19], a parameter of latent heating (*pht*) term is introduced to perform the modification with the model level $k$ as the variable.

$$mpten(k) = mpten(k) \times pht(k) \tag{1}$$

A Gaussian-form *pht* term is applied vertically as a function of the model level k and is fixed in time:

$$pht(k) = B + \frac{A}{\sigma\sqrt{2\pi}}e^{-\frac{(k-k_0)^2}{2\sigma^2}} \tag{2}$$

where, $B$ is the baseline parameter, which is set to 0.7 to eliminate overestimation of convection by the model; $k_0$ is the mean level that denotes the center height of the function; and $A$ and $\sigma$ are the amplitude and variance, respectively. In the research of Palmer et al. [16], the perturbation was randomly generated all over the region with certain horizontal scales corresponding to the equivalent of a Gaussian form. Their perturbation also had a spatial autocorrelation in the vertical direction, but the scale was only related to the horizontal perturbation. Therefore, their random perturbation was not very consistent with the ongoing convection system. In this study, the *mpten* perturbation is expressed in a nonrandom multiplicative form, so it is most obvious where the original large *mpten* exists, making it able to follow the convection structure and reveal its effect on different heights. The perturbation terms of [16,19] were set within the scales of [0.5 and 1.5] and [0.1 and 2.0], respectively, from the original tendency. In this paper, groups of experiments are designed and carried out, as shown in Table 1 and Figure 4, with CNTL serving as the unmodified simulation. The overall *pht* values range within [0.7 and 1.4] for HTK12/14/16 and within [0.7 and 2.0] for HTK8/10/1012. The amplitude is specifically set to be larger for the lower-level experiments, as the original LH is lower at lower levels than that of mid-to-upper levels (as indicated later in Section 4.2). The HTK8–16 experiments test the effects of amplified heating at different heights. Particularly, experiments HTK8, HTK10, HTK12, HTK14, and HTK16 test the heating effects at heights of 0–2 km (with maximum amplification center at about 1.5 km), 1.5–3.5 km (2.5 km), 2–5 km (3.5 km), 3–7 km (5 km), and 4–8 km (6.5 km), respectively (Figure 4). In this way, we are able to perturb the LH height without having to consider detailed processes inside the microphysics scheme or the model, making the perturbation mostly within the convection region where LH is most significant.

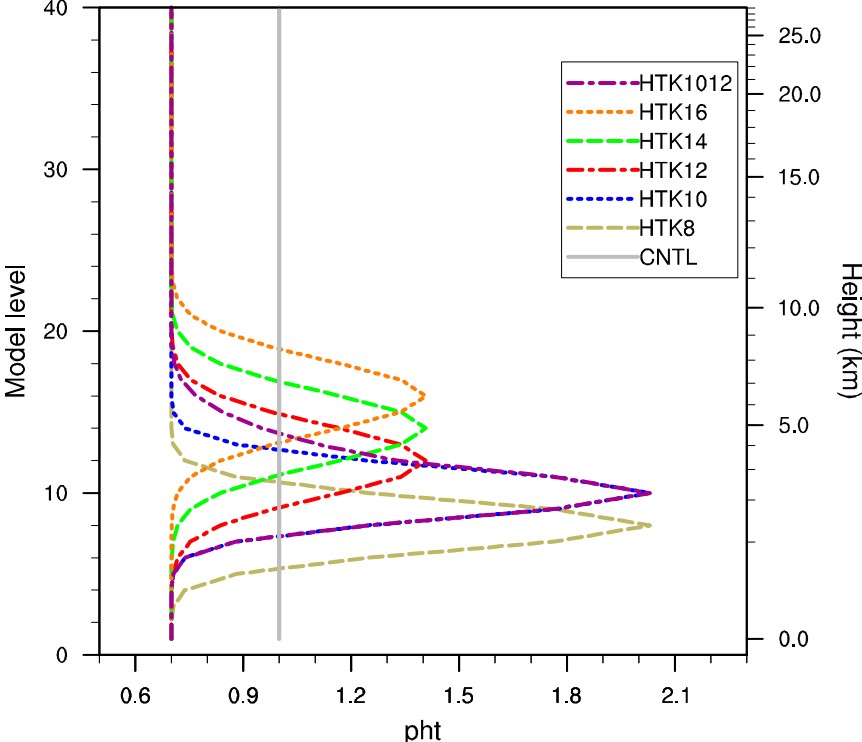

**Figure 4.** Profiles of the *pht* term for each experiment.

**Table 1.** Experimental design: *pht* coefficients used for each experiment.

| *pht* Coff  Experiment | B | A | σ | $k_0$ |
|---|---|---|---|---|
| CNTL | 1.0 | 0.0 | - | - |
| HTK8 | 0.7 | 5.0 | 1.5 | 8 |
| HTK10 | 0.7 | 5.0 | 1.5 | 10 |
| HTK12 | 0.7 | 3.9 | 2.2 | 12 |
| HTK14 | 0.7 | 3.9 | 2.2 | 14 |
| HTK16 | 0.7 | 3.9 | 2.2 | 16 |
| HTK1012 | Blending HTK10 and HTK12. See Equation (3). | | | |

For HTK1012, the *pht* term is given within a wider vertical range:

$$pht(k) = \begin{cases} B + \dfrac{A_1}{\sigma_1\sqrt{2\pi}}e^{-\frac{(k-10)^2}{2\sigma^2}}, k < 12 \\ B + \left( \dfrac{A_1}{\sigma_1\sqrt{2\pi}}e^{-\frac{(k-10)^2}{2\sigma_1^2}} + \dfrac{A_2}{\sigma_2\sqrt{2\pi}}e^{-\frac{(k-12)^2}{2\sigma_2^2}} \right) \cdot 0.5, k \geq 12 \end{cases} \tag{3}$$

where, $B = 0.7$, $A_1 = 5.0$, $A_2 = 3.9$, $\sigma_1 = 1.5$, and $\sigma_2 = 2.2$.

## 4. Simulation Results

### 4.1. Convection Evolution

To give a general description of all the simulations, Figure 5 shows the squall line evolutions with subjectively determined isochrones along with composite radar reflectivity at 01:00 UTC on April 19 when the observed convection system reached its mature stage as a bow-shaped squall line about 360 km in length. A quantitative evaluation is also given in Table 2 of the squall line length, moving speed, and correlation along with the observation.

In the CNTL experiment, convections were isolated as eastern part and western part at 01:00 UTC on April 19. The eastern echoes were unorganized and moved slowly at about $14 \text{ ms}^{-1}$. The western echoes were stronger and more linearly organized. They strengthened and caught up with the eastern echoes after 02:00 UTC. The two parts consolidated into one bow-shaped squall line at 03:00 UTC, but the whole system fell way behind the observed squall line. HTK16 produced similar isolated, slow-moving convections as CNTL (Figure 5f). Convections in HTK14 were more consolidated, but the moving speed (~$15 \text{ ms}^{-1}$) was still slow compared to the observation (~$21 \text{ ms}^{-1}$).

The experiments of HTK10/12/1012 produced a longer and faster-moving squall line more similar to the observation. Convections were more consolidated into one quasilinear-shaped system at 01:00 UTC in these experiments with lengths over 230 km and moving speeds over $17 \text{ ms}^{-1}$. HTK10/1012 produced a bow-shaped structure as in the observation. Although the western part echoes still emerged in HTK10/12 (the gray dashed lines in Figure 5c,d), they were relatively weak and compensated for the southern part of the primary squall line. Of all the experiments, HTK1012 had the best simulation, producing a bow-shaped squall line of about 330 km in length with a moving speed of about $19 \text{ ms}^{-1}$.

**Table 2.** A summary of simulated squall line characteristics of line length at 01:00 UTC April 19 ($L_s$), moving speed at 22:00 UTC on April 18 to 01:00 UTC on April 19 ($V_s$) and correlation with the observed reflectivity at 01:00 UTC April 19 (*Corr*).

| Experiment | CNTL | HTK8 | HTK10 | HTK12 | HTK14 | HTK16 | HTK1012 | OBS |
|---|---|---|---|---|---|---|---|---|
| $L_s$ (km) | 163 | - | 235 | 242 | 262 | 150 | 332 | 358 |
| $V_s$ (ms$^{-1}$) | 14 | - | 17 | 18 | 15 | 12 | 19 | 21 |
| *Corr* | 0.636 | 0.668 | 0.666 | 0.694 | 0.680 | 0.642 | 0.718 | 1.0 |

Note: The squall line length $L_s$ is calculated from the black isochrones shown in Figure 5. The moving speed is calculated through maximum curvature locations near the middle part of squall lines. The correlation is calculated within the area of (114–118° E, and 21–25° N).

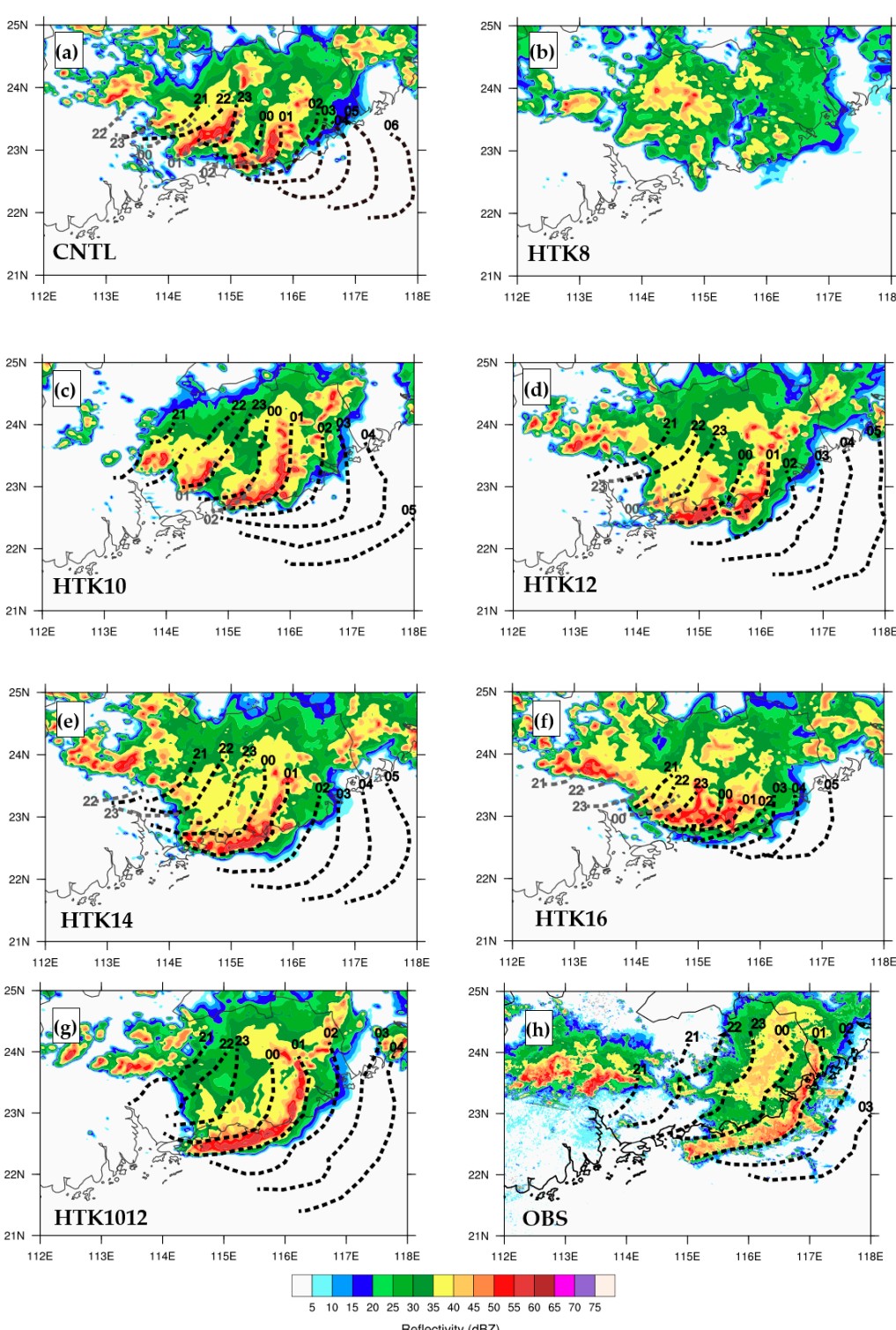

**Figure 5.** Radar reflectivity composited at 0–5 km (shaded) at 01:00 UTC on April 19 for (**a**) CNTL, (**b**) HTK8, (**c**) HTK10, (**d**) HTK12, (**e**) HTK14, (**f**) HTK16, (**g**) HTK1012, (**h**) Observation. The black/gray dashed lines are the isochrones of the squall lines subjectively determined (the black lines depict the main eastern part squall line; the gray lines depict the western part in some of the experiments).

Figures 6 and 7 compares composite radar reflectivity and 10 m wind among OBS, CNTL, HTK10, HTK12, and HTK1012 at the initiation and developing stages of the convection system. The 1 h temperature drop was also displayed to indicate cooling caused by

the fast-moving leading edge of the cold pool as in [40]. In observation (Figure 6a,b), radar echo began to exhibit some linear pattern at 21:00 UTC on April 18. A temperature drop of approximately −3 °C occurred behind the leading convection. At 23:00 UTC, the linear echo moved southeasterly and passed Huizhou (blue dot) with strengthened leading-edge cooling. Isolated new cells formed northwest behind the leading edge of the cold pool, caught up with the front part, and maintained their strength as they formed the southern part of the squall line. No obvious temperature drop was observed in conjunction with the northeast echoes (the echoes near Zijin), but the two parts of echoes combined to show a LEWP pattern.

The CNTL experiment (Figure 6c,d) failed to produce either the linear echo or the leading-edge cooling of the cold pool at similar scales as those found in the observations before 23:00 UTC on April 18. Although CNTL achieved maximum reflectivity of over 50 dBZ, the simulated convections remained separated and disorganized at 23:00 UTC. In HTK10 (Figure 6e,f), cold-pool cooling began to emerge near Guangzhou early, at 21:00 UTC. At 23:00 UTC, a more organized linear echo formed, and the leading edge of the cold pool in the southern part of the line moved through Huizhou with a weaker intensity than observation. In HTK12 (Figure 6g,h), cold-pool cooling strength was between HTK10 and CNTL at 21:00 UTC. At 23:00 UTC, the leading echoes showed a LEWP pattern similar to that seen in the observation, but the western echoes behind were stronger than the observed ones. In HTK1012 (Figure 7a,b), the convection echoes showed a linear pattern early, at 21:00 UTC. By 23:00 UTC, the convections rapidly propagated southeasterly with leading-edge cooling at a strength and size similar to the observation. Overall, HTK1012 produced convection evolvement that mostly resembled the observation at the initiation and developing stages of the squall line. Both HTK10 and HTK12 produced linear-shaped eastern echoes, but echoes in HTK10 were better organized, and they produced a better bow-shaped structure at 01:00 UTC on April 19. These early stage differences provided clues to the different convection patterns described above in Figure 5, and will be further analyzed later.

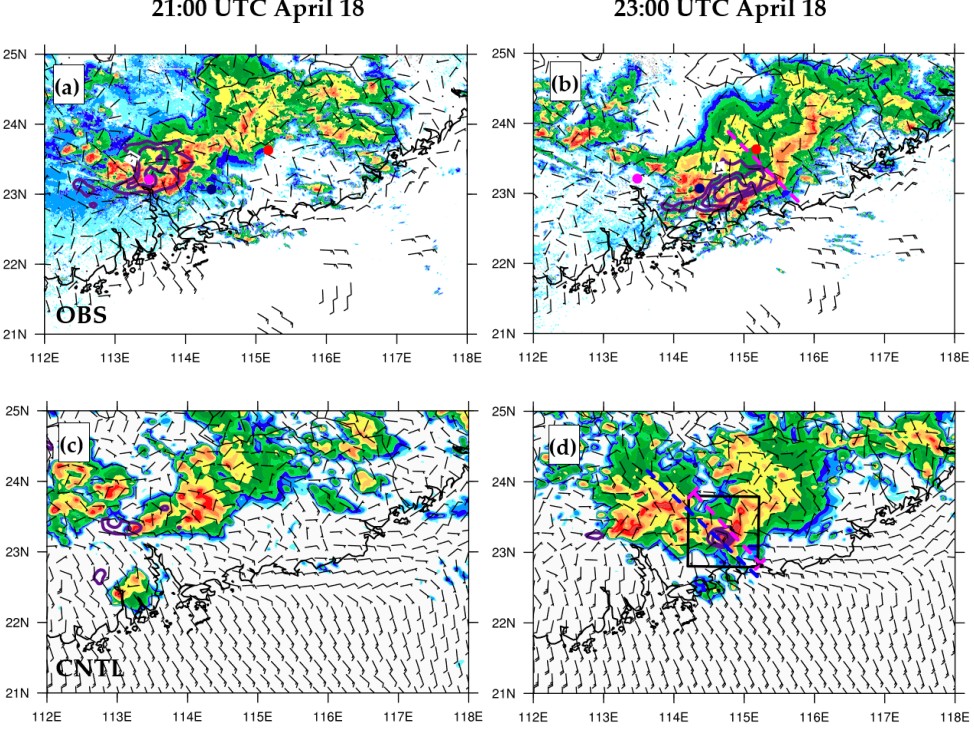

**Figure 6.** *Cont.*

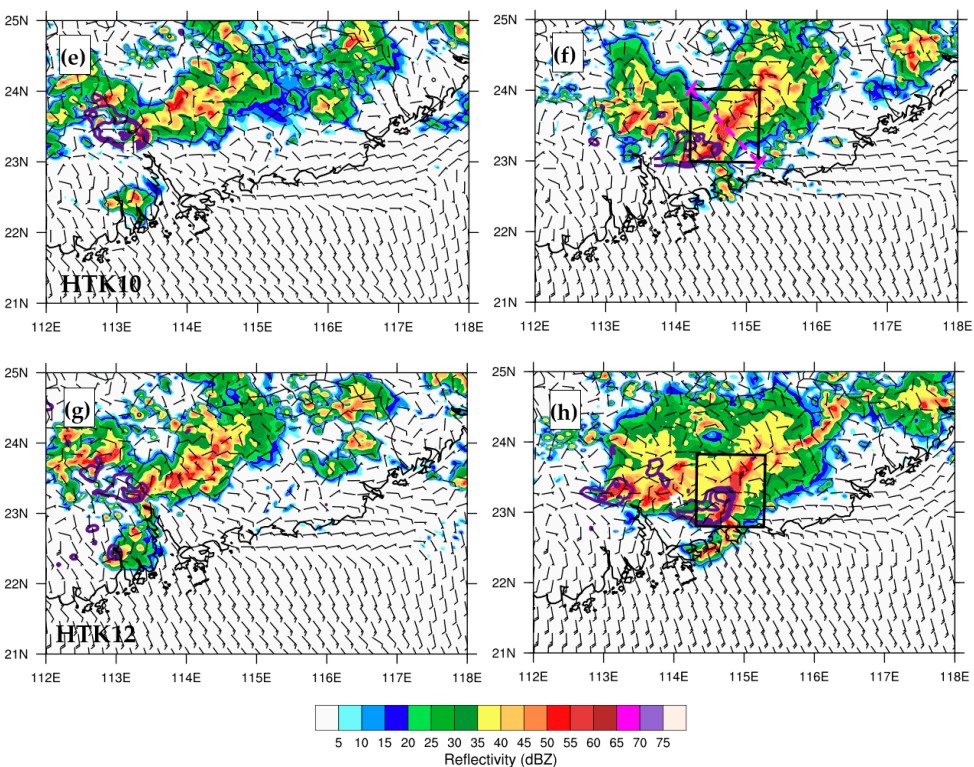

**Figure 6.** Radar reflectivity composited at 0–5 km (shaded), 2 m temperature difference in 1 h (deep blue contours starting from −1 °C at −1 °C intervals) and 10 m wind speed (wind barb, a full barb is 4 ms$^{-1}$) at 21:00 UTC and 23:00 UTC on April 18. (**a**,**b**) Observation. (**c**,**d**) CNTL. (**e**,**f**) HTK10. (**g**,**h**) HTK12. The magenta dots denote the station of Guangzhou, the deep blue dots denote the station of Huizhou, and the red dots denote the station of Zijin in (**a**,**b**).

**21:00 UTC April 18**　　　　**23:00 UTC April 18**

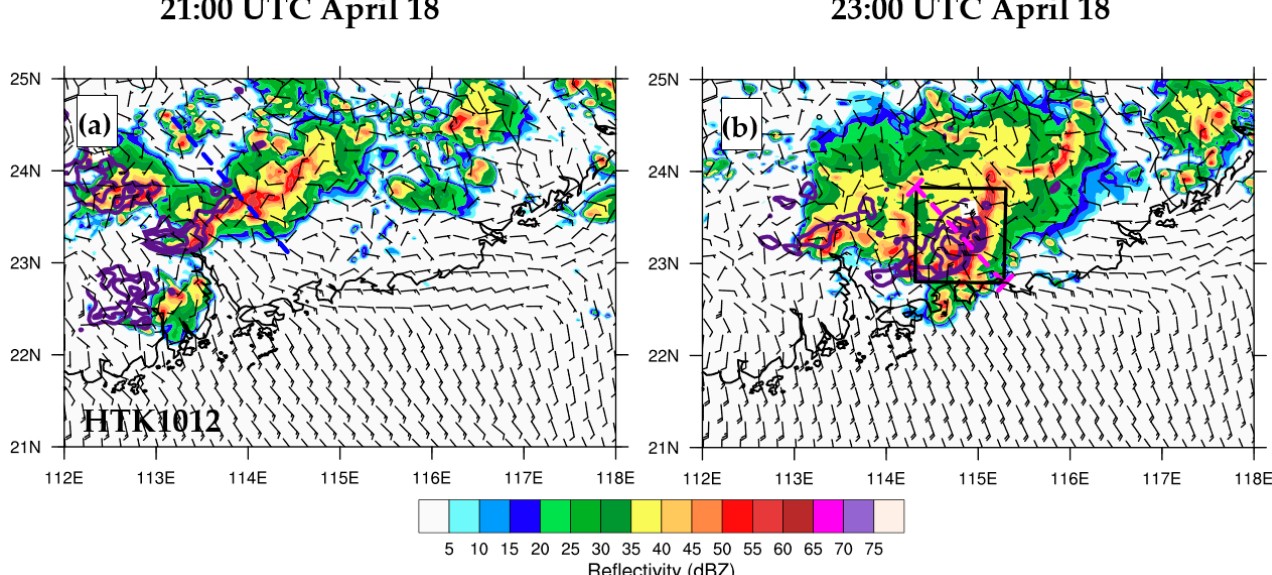

**Figure 7.** Same as Figure 6, but for experiment HTK1012. (**a**) 21:00 UTC. (**b**) 23:00 UTC.

*4.2. LH Profiles within the Convection System*

Figure 8 shows the different area-averaged heating amounts at 19:00 UTC and 23:00 UTC on April 18. The areas are subjectively chosen based on the evolution of the eastern part of the convection system in each experiment (some of these areas are denoted by rectangular boxes in Figures 6 and 7). From 19:00 UTC to 23:00 UTC, the strength of heating

inside the box almost doubles as the convection system evolves in time (Figure 8a,b, except HTK8, for which convection is inhibited by a lack of LH; the results of this experiment are excluded in the following discussion). Overall, the magnitudes or peaking heights of area-averaged LH among different experiments did not vary much, as shown in Figure 8a,b. Less LH is found in CNTL at 1–3 km compared to that seen in HTK10/12/14/1012 (Figure 8a,b). Average cooling near the surface was not obvious at 19:00 UTC (Figure 8a) during the initiation of convection and was mostly below 1 km at 23:00 UTC (Figure 8b).

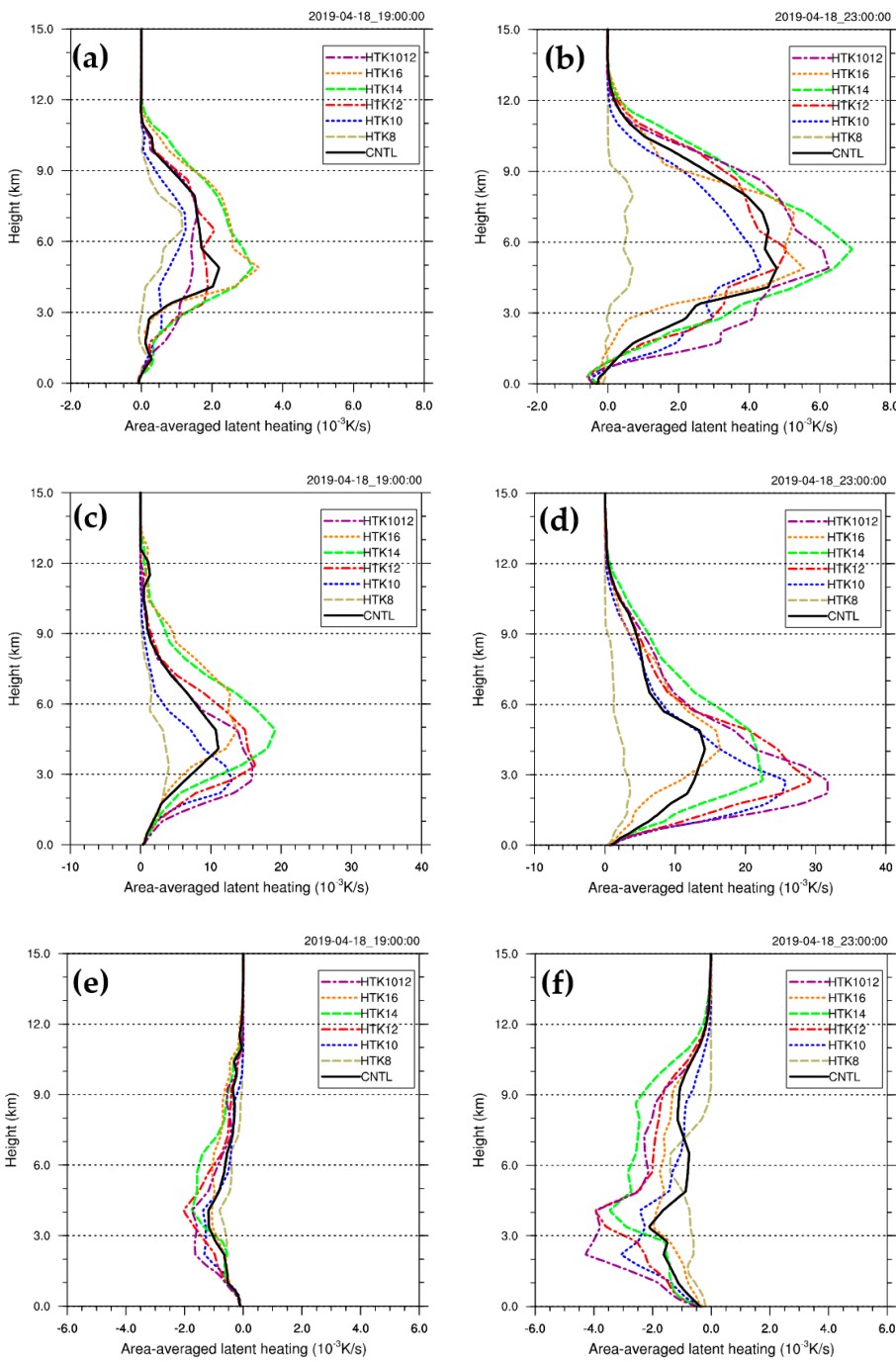

**Figure 8.** Area-averaged LH profiles for each experiment at 19:00 UTC (**a**,**c**,**e**) and 23:00 UTC (**b**,**d**,**f**) on April 18. The areas are subjectively chosen based on the evolution of reflectivity output in each experiment. Some of the areas are denoted by the square boxes in Figures 6 and 7. (**a**,**b**) Total averaged heating inside the box. (**c**,**d**) Averaged grids with positive heating inside the box. (**e**,**f**) Averaged grids with negative heating (cooling) inside the box.

At 19:00 UTC on April 18, 1 h after the model initiation, all experiments (CNTL/HTK10/HTK1012/HTK14/HTK16) exhibit similar composite reflectivity patterns (not shown), but the outputs start to differ in their LH profiles. The differences among experiments are clearer when the positive and negative LH grids are separated inside the box. The positive LH grids mainly represent the convection region of the system. As shown in Figure 8c, CNTL reveals a maximum heating center at about 4–5 km. HTK10, HTK12, and HTK1012 show maximum heating centers at about 3 km. HTK14 and HTK16 show maximum heating centers at about 5 km. These profile patterns reveal the effect of the *pht* modifications shown in Figure 4. The height of the maximum cooling center is basically the same for all the experiments, approximately 4 km. However, HTK10/HTK1012 has an extra cooling center located lower at about 2 km (Figure 8e).

Between 20:00 and 23:00 UTC on April 18, the convection systems continued to evolve, and the reflectivity patterns start to differ among the experiments. At 23:00 UTC, the magnitudes of total LH seem similar among the experiments (Figure 8b), but when we investigate the heating and cooling regions separately, the profiles of the experiments vary (Figure 8d,f). For the vertical distribution, CNTL outputs a maximum heating center remaining at about 4 km. HTK10/12/1012/14 have maximum heating centers located at about 2–3 km in height (Figure 8d). The heights of the maximum cooling centers for CNTL and HTK12/14/16 are at about 3–4 km. In HTK10/1012, the cooling centers descend lower to about 2–3 km (Figure 8f).

## 5. Convection Structures and Conceptual Model

### 5.1. Characteristic Flows during Storm Evolution

5.1.1. Rear-to-Front (RTF) Flows at Different Scales

The sensitivity experiments produce different convection patterns, and similarly different convection structures. In this case, the ambient inflow to the lower-front side of the squall line is similar among all experiments. Therefore, we first investigate the different flows behind the convection region among experiments. In this section, vertical cross sections are obtained from northwest to southeast to demonstrate convection evolution in both storm-relative and ground-relative ways. The term "rear-to-front (RTF) flow" is used for the ground-relative northwesterly flow from the backside of the convection system, while the term "rear inflow" is used for the storm-relative part of the extensive RTF flow.

Figure 9 shows the cross-plots resulting from CNTL/HTK10/HTK1012/OBS at 23:00 UTC on April 18 along the sections denoted in Figures 6 and 7. Before convection initiation, unstable air with high $\theta_e$ occupied the lower atmosphere with a depth of approximately 3 km (as indicated by the right part of the convection region, Figure 9a–c). Mid-level, low-$\theta_e$ air descended after convection formed (left part of Figure 9a–c). In HTK1012, more mid-level air with $\theta_e$ below 332 K descended to approximately 1.5 km above the surface behind the convection region. In CNTL, this air only descended slightly to a height of approximately 4 km. HTK10 shows results in between those of HTK1012 and CNTL. Although all simulations still produced stronger reflectivities, especially in the upper level, HTK1012 exhibits the broadest TS echo area at x < 90 km similar to the observation (Figure 9d).

Zhang and Gao [41] demonstrated that there were three components of a rear inflow relative to a TS squall line: a large-scale, deep, rear-to-front flow within the upper half of the atmosphere; a mesoscale-enhanced trailing inflow; and a latent-cooling- and water-loading-caused descending inflow near the cold-pool scale. In our case, two different-scale RTF flows are studied in the following part, one is the convective-scale descending flow near the convection area and the other is the mesoscale RTF flow in the mid-to-lower atmosphere behind the convection region. As shown in Figure 10a–c, all three experiments output vast mesoscale RTF flow areas over 100 km in size. Among them, HTK1012 shows the strongest RTF flow, at over 24 ms$^{-1}$, right behind the convection area, with its center at a height of approximately 1.5 km. The mesoscale RTF flow shown in HTK1012 overwhelms the descending flow near the convection area (Figure 10a). The mesoscale RTF flow revealed

in HTK10 is elevated, with its center at a height of approximately 3 km at x ≈ 0–60 km (Figure 10c). At x ≈ 60–90 km, the convective-scale descending flow is concentrated to the surface with a speed over 20 ms$^{-1}$ along the cross section. The mesoscale RTF flow revealed in CNTL is more elevated, with its center at a height of approximately 4–5 km (Figure 10b), while the descending convective scale RTF flow is also restricted to the surface as in HTK10.

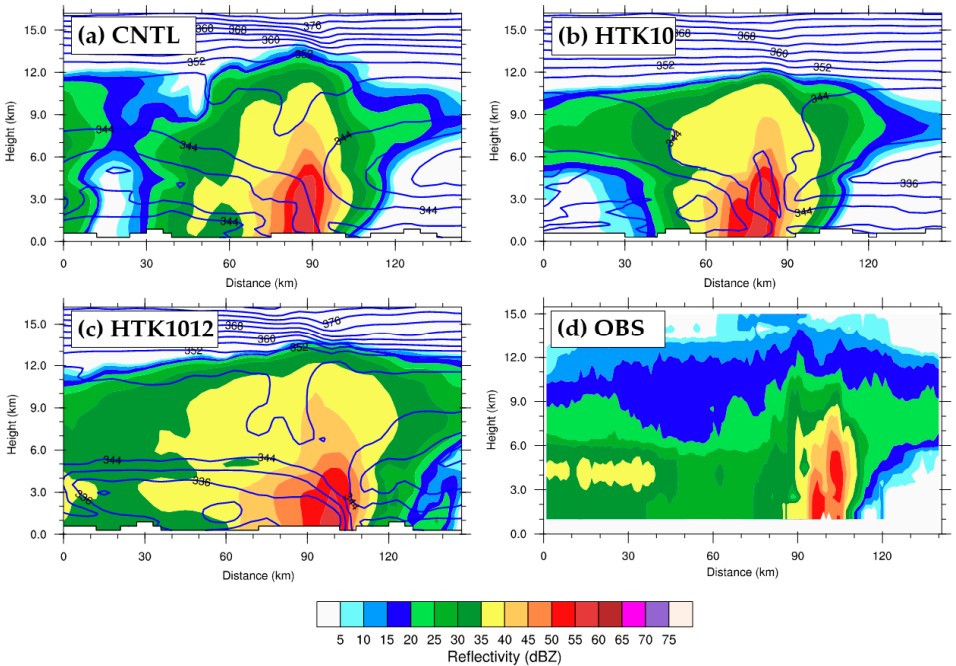

**Figure 9.** Vertical cross sections at 23:00 UTC on April 18 for HTK1012, CNTL and HTK10 along the dashed magenta lines denoted in Figures 6 and 7 with radar reflectivities (shaded) and equivalent potential temperatures (blue lines, contours at 4-K intervals).

In CNTL and HTK10, the convective scale RTF flow is separated from the mid-level mesoscale RTF flow located behind (Figure 10b,c). This is clearer in the storm-relative plots shown in Figure 10d–f, in which the rear inflow of each experiment is displayed. These plots are generated by subtracting the speed of the along-line system as defined by the gust front (which are approximately 17, 11, and 13 ms$^{-1}$ for HTK1012, CNTL, and HTK10, respectively). The strengths of the rear inflow near the convection area are over 8, 4, and 4 ms$^{-1}$ for HTK1012, CNTL, and HTK10 (Figure 10d–f), accounting for approximately 32%, 27%, and 24%, respectively, of the absolute strength of the RTF flows at the same location in Figure 10a–c. As discussed by Houze [1] and Pandya and Durran [6], coherent storm-relative rear inflows and FTR inflows exist when a rearward-tilted convection region heating core is considered. When unslanted vertical thermal forcing conditions were applied, only weak rear inflows in the lower level were observed (see Figure 20c of [6]). In our case, although different in strength, the heating in the convection region is almost vertical in all experiments, and the cooling lies lower behind the heating region (not shown), so the low-level storm-relative rear inflow occurs on a restricted scale of no more than 40 km (Figure 10d–f).

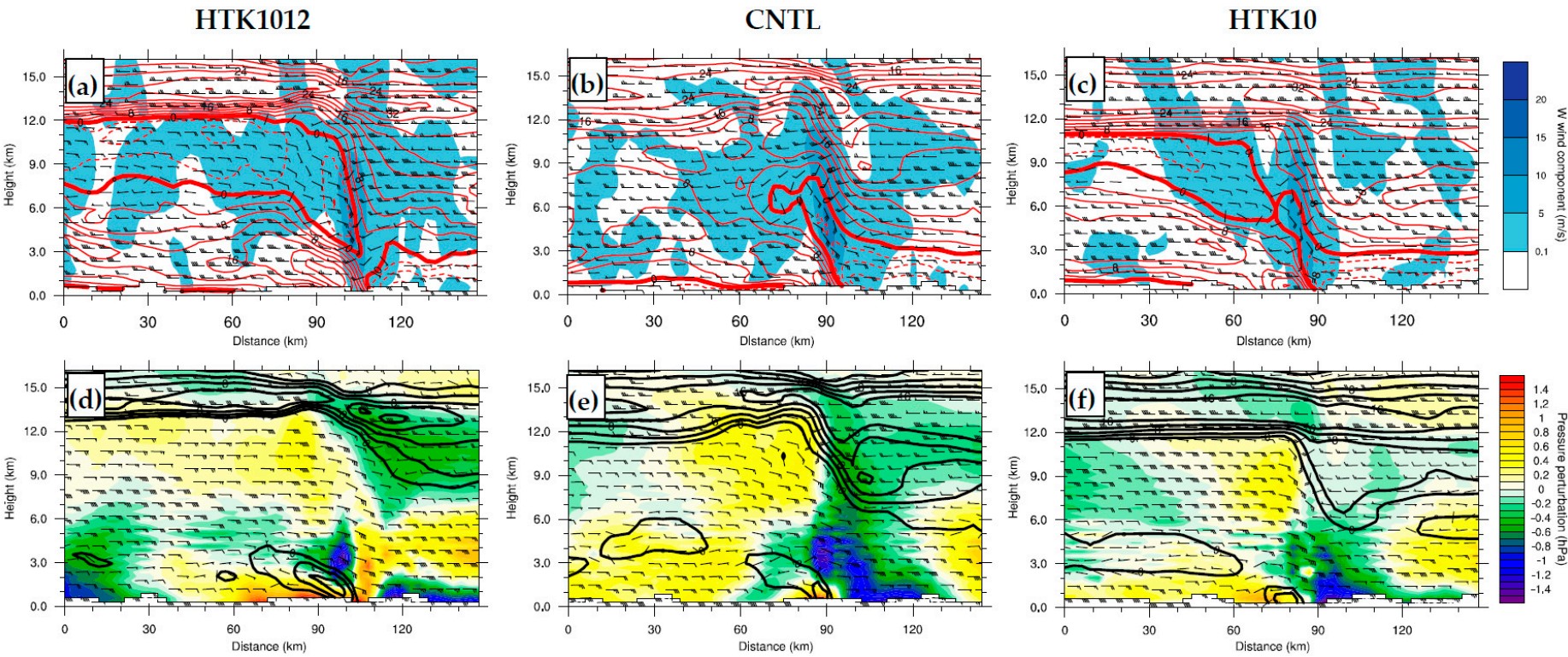

**Figure 10.** As in Figure 9, but for (**a**–**c**) horizontal wind along the cross sections (contours at 4-ms$^{-1}$ intervals, with negative values dashed), wind along the cross sections (wind barb, a full barb is 4 ms$^{-1}$), and vertical wind (shaded). (**d**–**f**) Horizontal pressure perturbations (shaded), relative wind along the cross sections (wind barb, a full barb is 4 ms$^{-1}$), and relative horizontal wind along the cross sections (black lines, contours range from 0 ms$^{-1}$ to 16 ms$^{-1}$ at 4-ms$^{-1}$ intervals). The quantities are shown as three-slice averages with approximately 0.2° width, as indicated by the segments at both ends of the dashed magenta lines in Figures 6d,f and 7b.

5.1.2. Differences in the Tilted Upward Flows

Differences in RTF flows among experiments are accompanied by different upward flows above them. As shown in Figure 10a–c, both HTK1012 and HTK10 revealed organized, tilted front-to-rear (FTR) flows, while in CNTL the upward flow mostly turned to the front side in the upper atmosphere.

As shown in Section 4.2, the experiments differ in their output convection heating heights due to the modification of the temperature tendency in the model. As LH and vertical wind are closely coupled in convections, such different heating heights can then cause the vertical wind core to occur at different heights. Figure 11 scattered the height and speed of maximum vertical wind at around 22:00 UTC and 23:00 UTC when convection was developing. Basically, there were more vertical wind maximums in CNTL in the mid-to-upper levels at about 6–11 km than that in HTK10 and HTK1012. At 22:00 UTC when convections were less developed, strong vertical winds over 10 ms$^{-1}$ were often located above 6 km height in CNTL, while in HTK10/HTK1012 they tended to locate at about 3–6 km height (Figure 11a). Convections intensified within the next hour for all the three experiments, and vertical winds over 15 ms$^{-1}$ appeared more frequently in CNTL and HTK10 below 6 km height (Figure 11b). However, there were still more markers at above 8 km in CNTL than those in HTK10/HTK1012, which indicated that the convection tended to keep strengthening while moving upward, causing its center to often be elevated to the mid-to-upper troposphere.

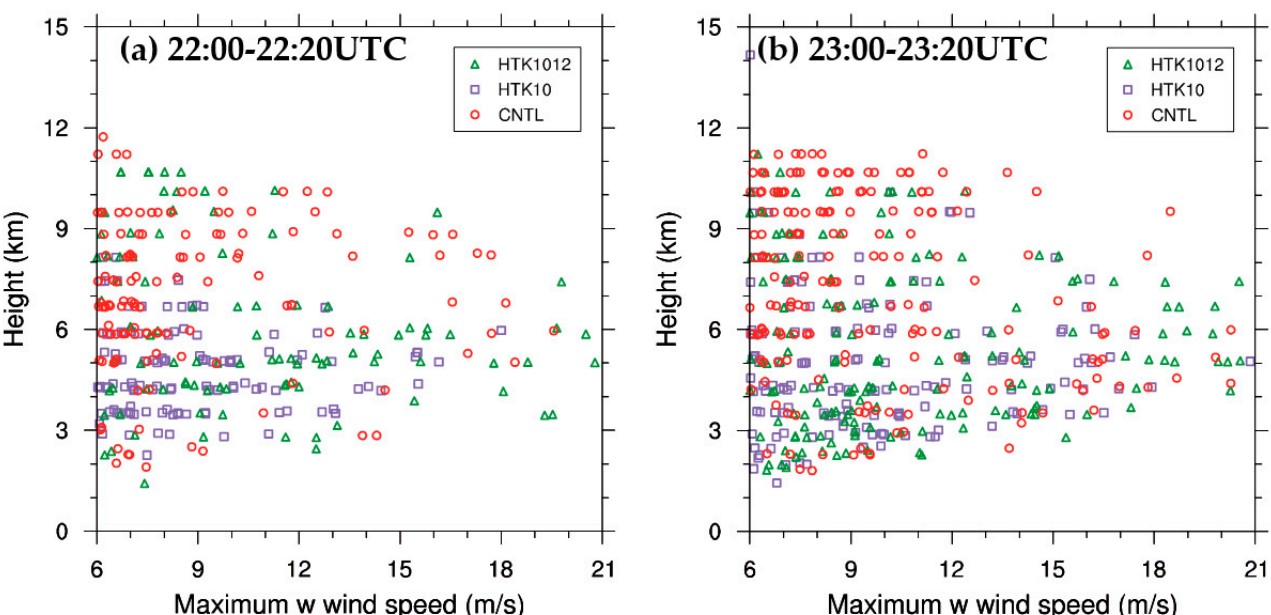

**Figure 11.** Scatter plot of maximum vertical wind and its height of each grid column in the 1° × 1° region of CNTL, HTK10 and HTK1012 at (**a**) 22:00–22:20 UTC, and (**b**) 23:00–23:20 UTC on April 18. Each experiment is calculated using data at 10 min interval within the square boxes in Figures 6 and 7. Only vertical speeds over 6 ms$^{-1}$ of are shown.

*5.2. Horizontal Vorticity, Cold Pool, and Conceptual Model*

As the RKW theory [42,43] indicated, different modes of convection could be induced by different strengths of cold-pool circulation and ambient shear indicated by horizontal vorticities. The convection tended to tilt rearward when the cold pool circulation to the backside of convection overwhelmed the ambient shear to the frontside and vice versa. In this section, detailed horizontal vorticity cross sections of CNTL and HTK1012 are described to explain the different convection structures among them (Figures 12 and 13). Cross sections for CNTL are taken at around 22:00–22:40 UTC on April 18 across the leading part of convection, while cross sections for HTK1012 are taken earlier at 20:20–21:50 UTC when the descending mesoscale RTF flow first starts to emerge.

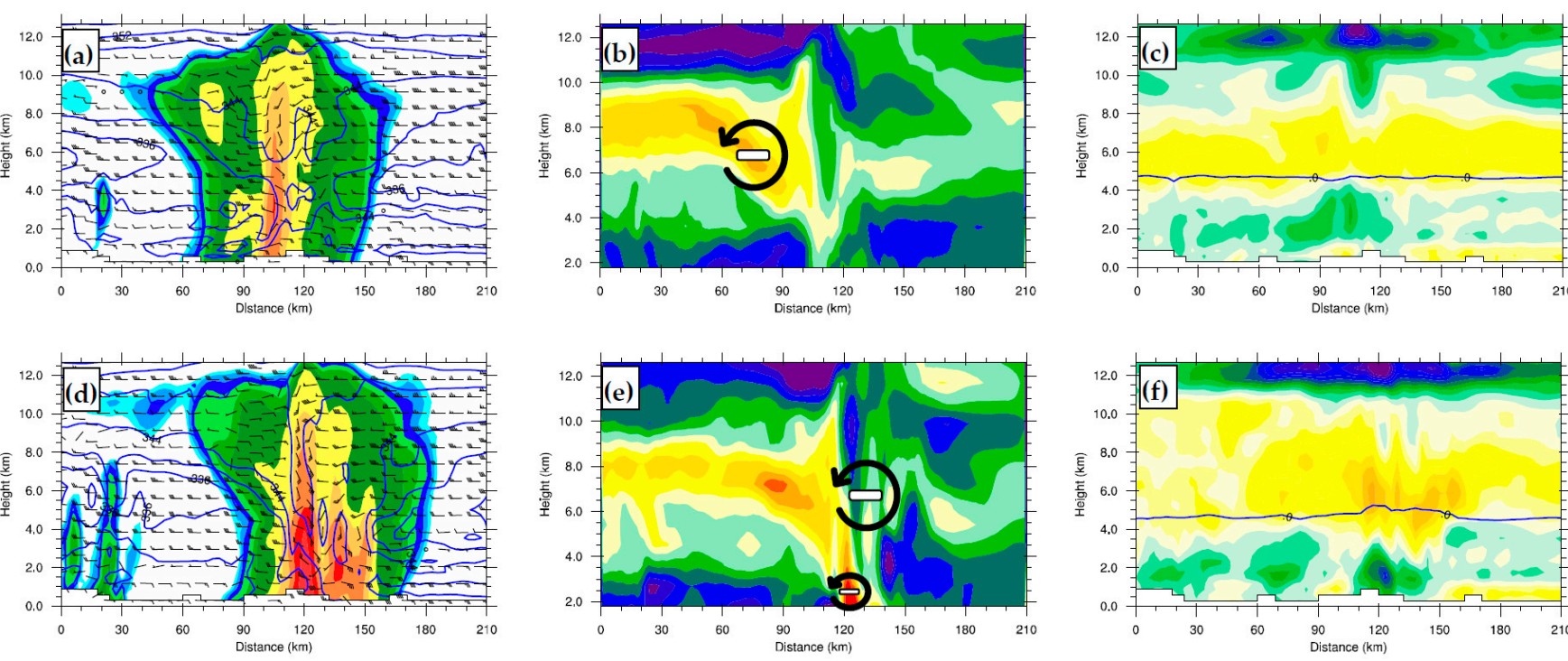

**Figure 12.** *Cont.*

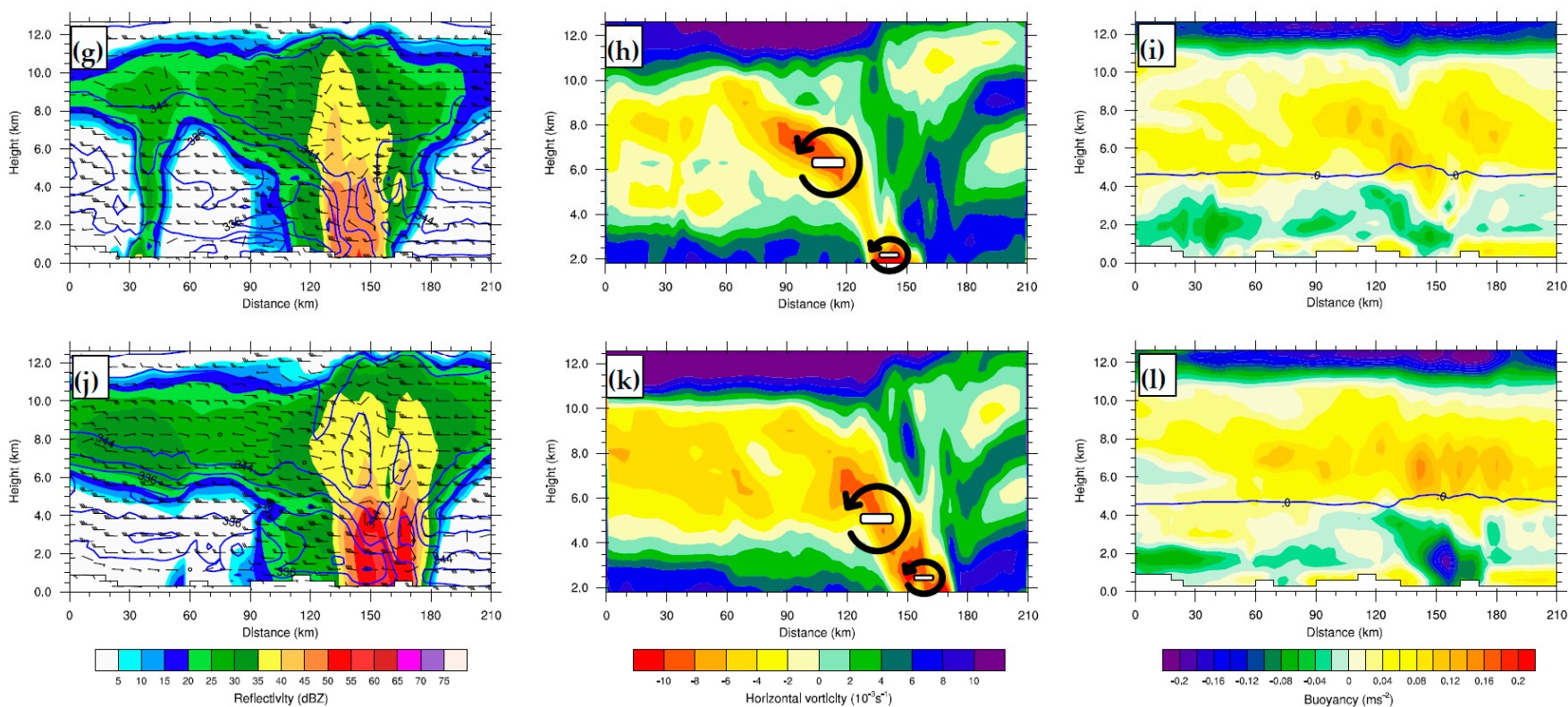

**Figure 12.** Vertical cross sections for HTK1012 along the dashed blue line denoted in Figure 7a at 20:20 UTC (**a–c**), 20:50 UTC (**d–f**), 21:20 UTC (**g–i**) and 21:50 UTC (**j–l**) on April 18. Left column: radar reflectivities (shaded), equivalent potential temperatures (blue lines, contours at 4-K intervals) and wind along the cross section (wind barb, full barb is 4 ms$^{-1}$). Middle column: horizontal vorticity (calculated with an interpolated 1 km × 1 km grid). Right column: buoyancy (shaded) and 0 °C isotherm (blue line).

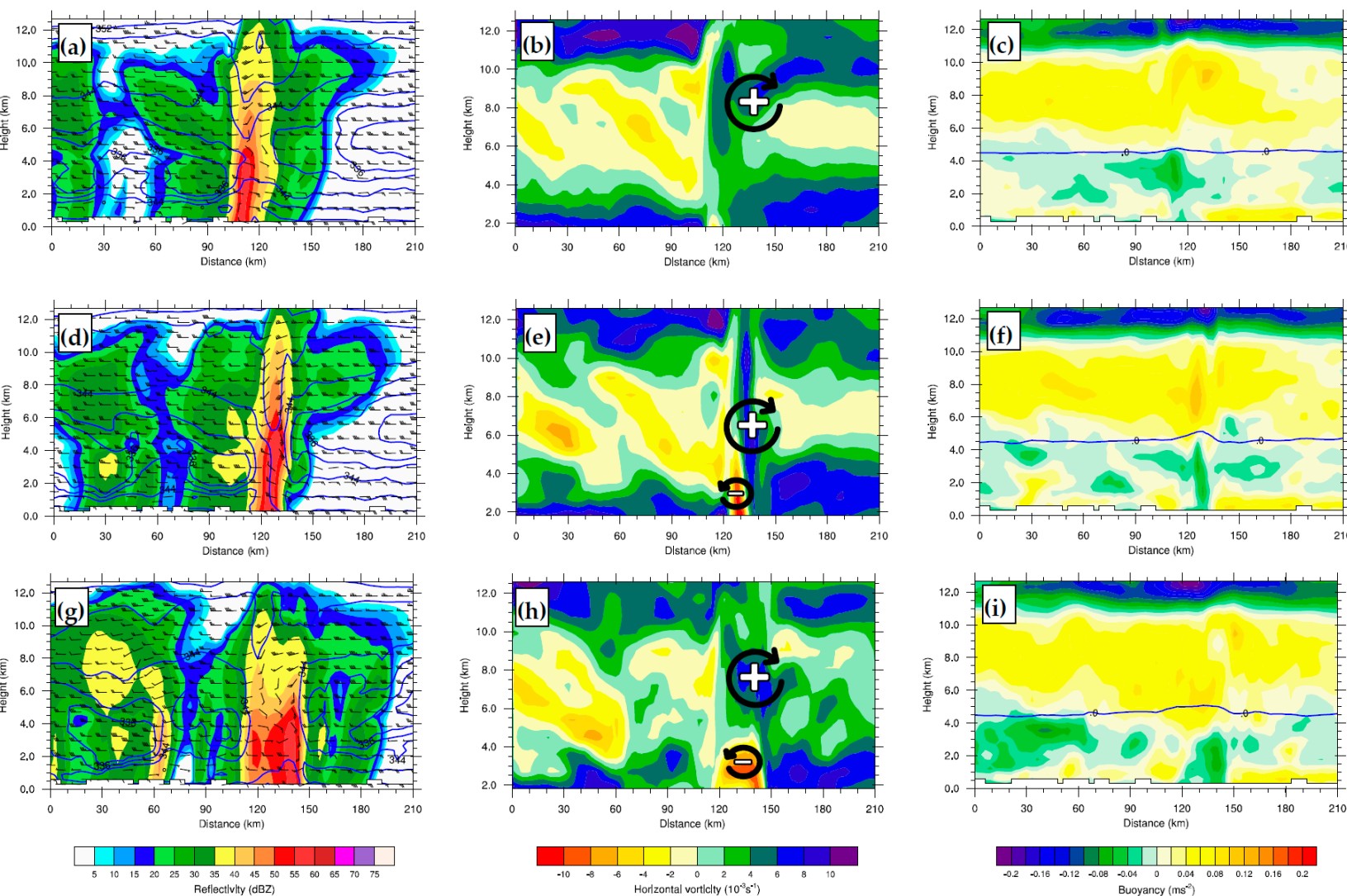

**Figure 13.** As in Figure 11, but for vertical cross sections for CNTL along the dashed blue line denoted in Figure 6d at 22:00 UTC (**a–c**), 22:20 UTC (**d–f**) and 22:40 UTC (**g–i**) on April 18.

At 20:20 UTC, early convection of HTK1012 is found at x ≈ 90–100 km, the system is overall a balanced type but starts to show some back-tilting TS pattern (Figure 12a) with a weak FTR flow at about 10 km height in the upper level behind the convection region (x < 100 km). A mesoscale RTF flow with $\theta_e$ < 340 K starts to descend from the mid-to-up level at x < 60 km to lower level near the convection region at x ≈ 100 km (Figure 12a). The descending part of RTF flow lies just beneath a mid-to-up level descending negative vorticity region at x ≈ 60–100 km, as shown in Figure 12b. This negative vorticity is composed of both the RTF flow and the weak FTR flow upon it. At 20:50 UTC, the upper-level FTR flow deepens at about 7–11 km at x ≈ 90–110 km, and the RTF flow continues to descend to near-surface at x ≈ 90–120 km (Figure 12c). In the vorticity section, the mid-to-up level descending negative vorticity region strengthens at x ≈ 60–110 km, with its front reaching down to the backside of the convection region. A sudden burst of near-surface negative horizontal vorticity appears just in front of the descending vorticity region at x ≈ 120–130 km. At 21:20–21:50 UTC, a coherent FTR tilted flow and a deep descending RTF flow with low $\theta_e$ formed in the vast region behind the convection region (Figure 12e,g). In the vorticity section, the descending negative vorticity region connects with the strong near-surface negative vorticity in the convection region (Figure 12f,h). Such flow structure sustains in the next few hours, making the squall line a TS form in HTK1012.

Unlike HTK1012, the front southeastern part convection in CNTL is more likely an LS type with RTF flow of $\theta_e$ < 340 K, about 2–6 km in height behind the convection region and no obvious FTR flow above it (Figure 13a,c,e). A mid-to-up level descending negative vorticity region is also found in CNTL at x ≈ 40–110 km at 22:00 UTC (Figure 13b) but with a weaker strength compared to HTK1012 (Figure 12b). Meanwhile, CNTL has a much stronger mid-to-up level positive vorticity region to the front side of the convection region than HTK1012 (Figure 13b,d,f), which means the convection upward flow tends to turn frontward when coming up. Notably, a sudden burst of near-surface negative horizontal vorticity occurs in front of the descending vorticity region (Figure 13d) similar to that of HTK1012 (Figure 12d), but as the negative vorticity at x ≈ 60–120 km weakens at 22:40 UTC, the near-surface convective-scale vorticity loses its supply and starts to decay (Figure 13f). At this time, a stronger descending vorticity region appears at x ≈ 0–60 km behind the front part convection, and later the northwestern part convection in CNTL develops in front of it.

One question here is, what is the relation of the mid-level FTR flow to the cold pool below? As indicated by previous studies [42,43], the near-surface negative vorticity is caused by buoyancy gradients along the leading edge of the cold pool. Meanwhile, the moving speed of the system is also related to the cold pool strength. The buoyancy *B* and theoretical speed of cold pool *C* are defined following [42,44]:

$$B = g\left[\frac{(\theta - \bar{\theta})}{\bar{\theta}} + 0.61(q_v - \bar{q_v}) - q_t\right] \tag{4}$$

where, $\theta$ is potential temperature, $q_v$ is water vapor mixing ratio, and $q_t$ is total condensate mixing ratio. $\bar{\theta}$ and $\bar{q_v}$ are horizontally averaged potential temperature and water vapor mixing ratio, respectively. In this study, we use the profiles at the initial time of 18:00 UTC on April 18 to calculate the average term similar to the study of Xue et al. [44].

$$C^2 = 2\int_0^H (-\bar{B})dz \tag{5}$$

where, *H* is the cold pool depth, which we set to 4 km in this study; and $\bar{B}$ is the horizontally averaged buoyancy within the cold pool, which is calculated using negative buoyancy grids within the region of the gust front and 30 km behind it.

In HTK1012, no obvious cold pool was found before 20:20 UTC on April 18. The negative buoyancy region at x ≈ 60–120 km was mostly elevated at about 1–4 km height below the melting layer (Figure 12c). At 20:40–20:50 UTC, the cold pool indicated by negative buoyancy reaching to surface was found below the convection region for the

first time (Figure 12f). The cold pool weakened after 21:00 UTC as it moved fast forward (Figure 12i), strengthened again after 21:40 UTC, and maintained since then. Although the cold pool was generated later in CNTL than that in HTK1012, negative buoyancy reaching to surface was still found below the convection region after 22:10 UTC (Figure 13f,i). When we compare the evolution of minimum horizontal vorticity behind the cold pool and cold pool speed in HTK1012 (Figure 14a), it is found that the peaking of cold pool speed usually came (at 20:40/21:50 UTC) right after the minimum descending of mid-to-lower level negative vorticity (blue arrows) in time. Noticing that in HTK1012, the mid-level FTR flow emerged at about 20:20–20:30 UTC (Figure 12a) before the emergence of surface cold pool at about 20:40 UTC, the descending of mid-to-lower level vorticity should favor the strengthening of cold pool as it introduced low-$\theta_e$ air from mid-level down to the convection region.

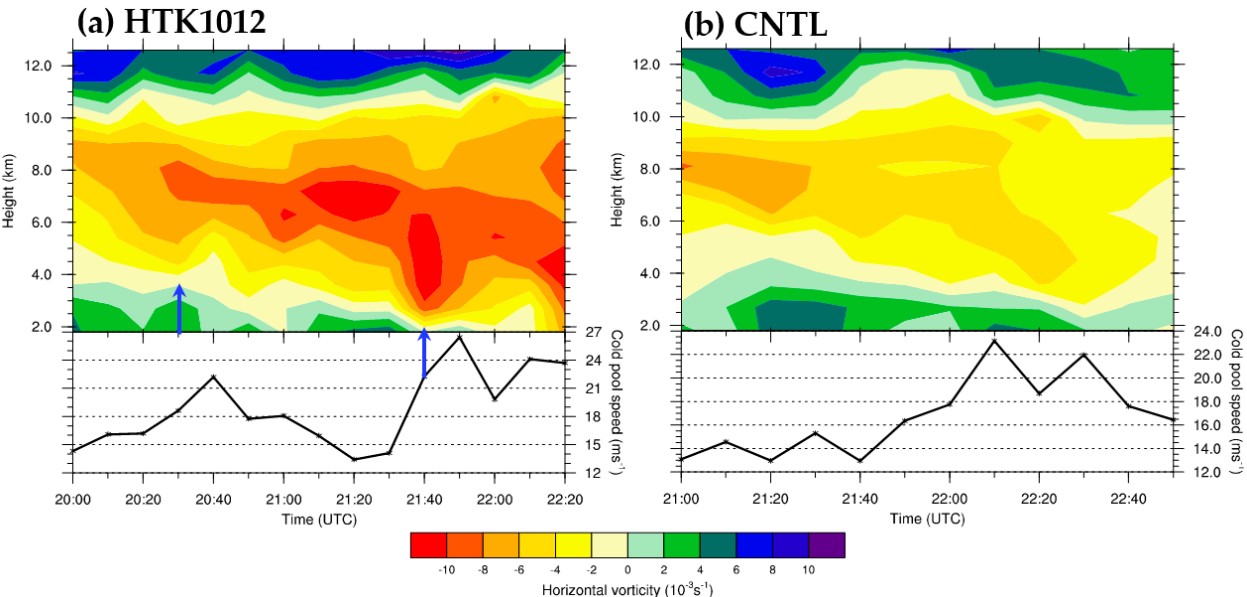

**Figure 14.** Height-time evolution of minimum horizontal vorticity (shaded) behind the cold pool along cross-sections of Figures 11 and 12. (**a**) HTK1012, 20:00–22:20 UTC April 18. (**b**) CNTL, 21:00–22:50 UTC April 18. Below is theoretical cold pool speed (black line) in the same time range.

Traditionally, the ratio of theoretical cold pool speed $C$ and the environment wind shear $dU$ is used to determine the balance between the voticities with the cold pool and the front environment. Figure 14 shows the evolution of minimum horizontal vorticity behind the cold pool along with the cold pool speed $C$. The cold pool speed is averaged within the region between the gust front and 30 km behind it. The minimum vorticity is calculated within the region between about 90 and 30 km behind the gust front to exclude the near-surface part vorticity related to cold pool circulation. In this case, the maximum $dU$ in the front environment between different levels from 0 to 4 km along the cross section is about 20 ms$^{-1}$. However, cold pool speed did not exceed 20 ms$^{-1}$ most of the time within 20:00–21:30 UTC (Figure 14a) when mid-level FTR flow already occurred in HTK1012. On the other hand, cold pool speed exceeded 20 ms$^{-1}$ after 22:10 UTC (Figure 14b) in CNTL, while no mid-level FTR flow was found around that time, only limited shallow FTR flow was found below 5 km near the cold pool top (e.g., Figure 13a,g).

The balance of a deep tilted FTR flow may be better understood by looking at the relative difference in horizontal vorticities at both sides of the convection (Figure 15). Here, the horizontal vorticity at each time is given by calculating the difference of the x-averaged negative vorticity behind the convection and x-averaged positive vorticity in front of the convection, both within about 60 km distance. In HTK1012, there was stronger negative vorticity in the mid-to-upper level behind the convection than the positive vorticity in front

of it all the time between 20:00 UTC to 21:50 UTC (Figure 15a). The near-surface negative vorticity related to the cold pool overtook the front environment shear after 20:30 UTC, then a coherent FTR formed through the near-surface to the upper level of the troposphere. There was stronger positive vorticity near the height of 4 km, which indicated stronger environment shear at a level little higher than the cold pool. On the other hand, positive vorticity overtook negative vorticity in the mid-to-upper level most of the time within 21:40 UTC to 22:50 UTC in CNTL, and although there was negative vorticity at near surface after 22:10 UTC, the tilted FTR flow only occurred within restricted region near the front edge of cold pool (Figure 13a,d,g). Note that negative vorticity was found in the mid-to-upper level at 21:00 UTC to 21:40 UTC in CNTL (Figure 15b); this was mainly caused by the vorticity from another cell (which dissipated later) behind the cell in concern.

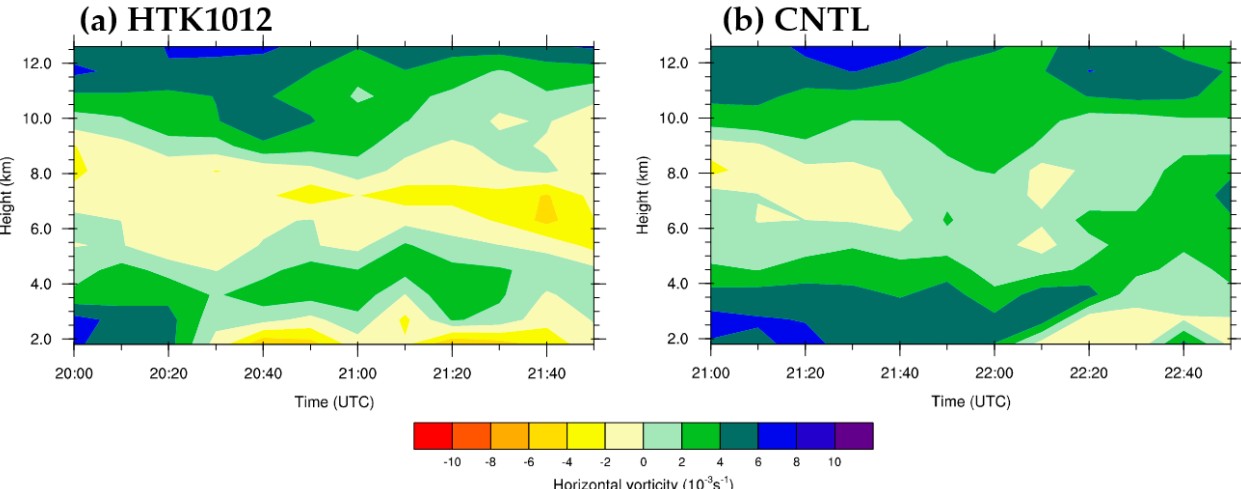

**Figure 15.** Height-time evolution of the difference of averaged horizontal vorticity in front/behind the squall line along cross-sections of Figures 11 and 12. (**a**) HTK1012 at 20:00–21:50 UTC, (**b**) CNTL at 21:00–22:50 UTC on April 18.

In this case, when the storm is within a mid-level environmental RTF flow, the classical RKW theory can be used by adding the effect of vertical winds centered at different heights (Figure 16). Traditionally, the tilt of the upward flow is determined by the relative strength of the cold pool circulation ($C_c$) and the environment shear circulation ($C_s$). In our case, given an upward environment wind shear composed of easterly flow below and westerly flow in the mid-upper troposphere, the tilting of the FTR flow (the thick, double-lined gray/green arrow) is related to circulations both at low-level and mid-level, thus affected by the height of the vertical wind core (the gray/green arrow) in the mid-to-low level atmosphere. When a lower-centered vertical wind core (gray arrow) occurs, it tends to enhance a lower mid-level circulation ($C_r$) by enhancing negative horizontal vorticity to the left side of the w wind core. In an unstable environment with easterly wind shear, such a lower $C_r$ favors a ground-relative FTR flow to the upper part of it. Beneath $C_r$ lies the mesoscale RTF flow, which can introduce mid-level low $\theta_e$ air down to the near-surface behind the convection. With the aid of microphysical cooling, it strengthens the cold pool circulation $C_c$ near the surface. The low-level positive environment shear circulation $C_s$ usually lies a little higher than $C_c$ in front of the convection. Such disposition of vorticity makes the inflow air first tilt rearward due to stronger cold pool circulation $C_c$, then bend to the downshear direction near the top of the cold pool due to $C_s$, and turns rearward again when it comes higher to the level near $C_r$. At the mature stage of the convection system, the mid-to-low level mesoscale circulation $C_r$ connected with the cold pool circulation $C_c$, favoring a coherent descending RTF flow (thick gray vector) and a rearward-tilted FTR upward flow (double-lined gray vector).

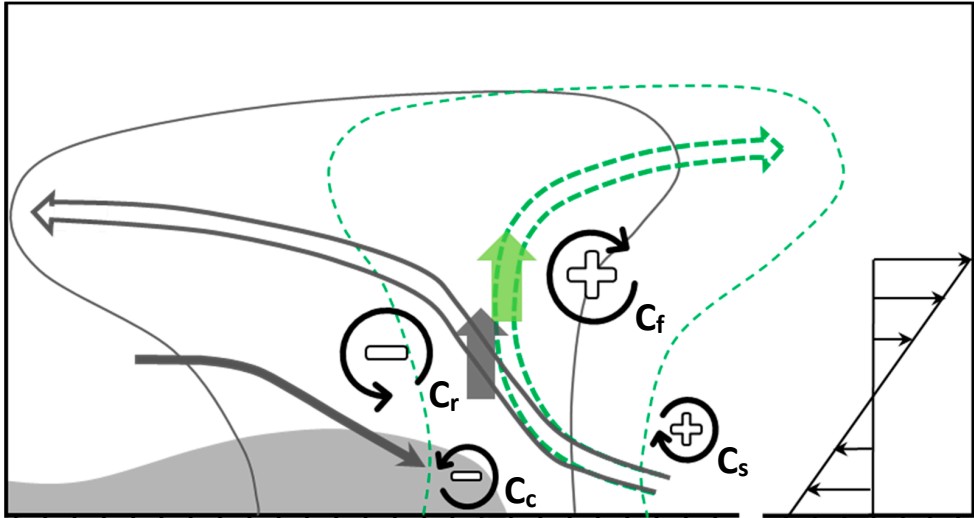

**Figure 16.** Conceptual model of a convective system superimposed by a lower/higher centered (denoted by a translucent gray/green arrow) updraft. The thin gray line indicates reflectivity boundary of a TS squall line. The thick gray vector and shading denotes the RTF descending flow and surface cold pool related to the TS squall line. The dashed thin green line indicates reflectivity boundary of a LS squall line.

On the other hand, when a higher-centered wind core (the green arrow) occurs, it tends to be advected to the front side of the storm by strong upper-level RTF flow in the environment, so to strengthen the mid-to-upper level circulation ($C_f$) to the up-front side of convection by enhancing the positive horizontal vorticity to the right side of the w wind core, thus favoring a frontward-tilted upward flow (dashed double-lined green vector).

## 6. Conclusions

In this study, the effects of different LH profiles on the structure and evolution of an intensive squall line in South China (18–19 April 2019) are investigated using CRM simulations. Although some previous studies found that CRMs tend to overestimate convection in the middle-to-upper troposphere, few related such biases to the height of LH profiles whose details were hard to determine using current observations. Here, we modify the model by applying a Gaussian-form multiplicative term (*pht*) to the latent heating term to alter the height level of LH, especially in the convection region and compare the results to the radar observation. The sensitivity experiments (HTK10/HTK12/HTK1012) conducted with amplified LH centered at a lower level than that in the CNTL experiment (peaking at around 2–3 km) simulate more organized and fast-moving squall lines, showing consistency with the observed TS squall line.

These HTK experiments conducted herein are related to a coherent RTF flow descending to the mid-to-low level behind the convection region. Results of this study indicate that the convection shear interaction should be studied to a deeper extent through the troposphere. In this case, three RTF flows of different scales were found to be similar with classical studies [41–43], while unlike traditional studies that attributed the tilted upward flow to the relative strength of low-level cold pool and ambient shear, we show that different structures of the RTF/FTR flows are also sensitive to different LH profiles at mid-to-low levels, and this may precede the low-level cold pool process; a deep tilted FTR upward flow should occur with stronger negative vorticity behind the convection than positive vorticity in front of it, both near the surface cold pool and at a broader region in the mid-level. Specifically, the rearward-tilted upward flow could be favored by a lower-centered vertical wind core by intensifying the mid-level circulation $C_r$ behind the convection region. A lower centered $C_r$ could introduce mid-level low $\theta_e$ air down lower to the near-surface of the convection region thus boosting the cold pool circulation $C_c$ there. The cold pool

intensified after the frontside of slanted mid-level negative vorticity region developed to lower level near the convection, and then boosted its speed forward. When the squall line reaches its mature stage with a TS form, a region of descending negative vorticity consisting of mid-level circulation $C_r$ and cold pool circulation $C_c$ together lies behind the convection region, favoring a coherent mesoscale RTF flow, which in turn contributes to both the strength and moving speed of the convection system.

In this study, the amplification of the heating is built in a multiplicative form, as in former studies, but not in a random way. Because such multiplicative modifications are performed at every time step in the model integration, these modifications are more evident in locations where large amounts of LH originate, following the movement of the convection system and perform systematic tuning on the height of LH profiles. In the ideal study of Pandya and Durran, it was found that the shape (slantwise/perpendicular) of the thermal forcing is crucial to the mesoscale circulation around the squall line [6]. However, in this study, the positive LH near the convection region is perpendicular-shaped among both the CNTL and sensitivity experiments; it is the vertical distribution of LH that varies. While direct tuning of LH heights improved the simulation, the fundamental source of such biases on heating heights remains to be found. Although we applied modifications to LH using the temperature tendency output from the microphysics schemes, it does not necessarily mean that such bias of LH comes from the schemes. At lower levels, LH profiles strongly depend on the vertical velocity in a saturated updraft to first-order approximation of $LH \sim -\rho w \cdot dq_s / dz$ (where $w$ is the vertical velocity and $q_s$ is the saturation mixing ratio). At above the melting level, studies have demonstrated that bias of LH was related to both the microphysics assumptions and unreal simulated updrafts [25,45]. The vertical velocity can be influenced by LH, but also by other factors, such as advection scheme, subgrid mixing, land-surface process, etc. Thus, the bias of LH may also attribute to these factors.

It should be noted that the results in this paper are based on a single squall line case, and more real MCS case studies are needed to improve and verify the results in future work. Modifications of the profiles of LH should be studied and achieved through more specified processes or schemes. Ideal simulations are also needed to help give a more quantitative explanation of the interaction among mid-level vorticity, cold pool, and ambient shear. Meanwhile, it is necessary to apply further observations in the convection regions of MCSs to investigate the detailed features related to LH, such as particle distribution and other thermodynamic elements. Observation experiments may be carried out by performing an intense sounding network surrounding certain MCS events to obtain the average LH profile there, so as to compare it with that from the numerical simulations. From a practical perspective, it is also possible to use more structured perturbations within the convection region in generating the ensemble forecast members to get a better dispersion of the numerical system.

**Author Contributions:** Conceptualization and methodology, H.C.; data assimilation, simulation and visualization, M.L.; formal analysis, H.C., X.B. and L.Z.; data curation, J.Z.; supervision, L.M. All authors have read and agreed to the published version of the manuscript.

**Funding:** This research was supported by the National Key Research and Development Project of China (Grant No. 2022YFC3003905), Research project of the Science and Technology Commission of Shanghai Municipality (Grant No. 19dz1200102), the Yangtze River Delta Science and Technology Innovation Community Field Project of Shanghai Science and Technology Innovation Action Plan (Grant No. 21002410200), and program of the National Natural Science Foundation of China (Grant No. 41975069).

**Acknowledgments:** The authors would like to thank Ming Hu of NCAR for the help in running WRF and GSI system, and Yanhong Wang of the Chinese Academy of Meteorological Sciences for the processing of the CinradMosaic system.

**Conflicts of Interest:** The authors declare no conflict of interest.

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
