# Peer review of "Impact of Explicitly Parameterized Mid-to-Low Level Latent Heating on the Simulation of a Squall Line in South China"

_water, doi:10.3390/w15091743_

Round 1
Reviewer 1 Report
Reviewer comment (R.C.): The Abstract section contains several grammatical errors, including missing articles, incorrect verb forms, and inconsistent use of tenses:
· The sentence "The impact of different LH profiles on the squall line is investigated by modifying the altitude of LH peaking at around 2-5 km" should be rewritten for clarity. A possible revision could be "We investigated the impact of modifying the altitude of LH peaking around 2-5 km on the squall line."
· "Increasing of LH" should be "increasing LH."
· The phrase "influencing the evolution of a front-to-rear tilted upward flow" is missing an article. It should be "influencing the evolution of the front-to-rear tilted upward flow."
· The sentence "The influence of different LH to the structure of simulated squall line is explained" should be revised to "The influence of different LH profiles on the structure of the simulated squall line is explained."
· "RKW theory consid- ering" should be "RKW theory considering."
· The sentence "Stronger LH at lower heights results in a lower-centered vertical wind core in the convection region, behind which in the mid-to-low level is a region of descending negative horizontal vorticity" should be revised for clarity. A possible revision could be "Stronger LH at lower heights results in a vertical wind core centered lower in the convection region. Behind the core at the mid-to-low level is a region of descending negative horizontal vorticity."
· "Descending flow beneath it" should be "Descending flow below it."
· "??" is a mathematical symbol and should be defined or replaced with words that explain its meaning in the text.
Reply:
Reviewer comment (R.C.): The authors need to improve the abstract. Therefore, the abstract should answer these questions about your manuscript: What was done? Why did you do it? What did you find? Why are these findings useful and important? Answering these questions lets readers know the most important points about your study and helps them decide whether they want to read the rest of the paper. Make sure you follow the proper journal manuscript formatting guidelines when preparing your abstract.
Reply:
Reviewer comment (R.C.): The introduction section needs improvement. It is important to clearly state the following aspects in the introduction:
· The main research question
· Hypothesis, if applicable
· Main objective of the study
· Gap in the literature that this study aims to address
· Motivation behind the research
· Relevance and significance of the topic
· Innovations and novel contributions of the study compared to existing work
· Unique contribution of this study to the field of investigation.
Reply:
Reviewer comment (R.C.): The introduction section contains several grammatical errors:
· In sentence 24, "proportion" should be followed by "of" instead of "in".
· In sentence 28, "especially in the vertical direction" should be followed by a comma.
· In sentence 29, "the other is through retrieval schemes using satellite remote sensing" should be followed by a comma.
· In sentence 36, "large-scale" should be hyphenated.
· In sentence 45, there seems to be a missing sentence or phrase between "real case" and "Previously" that could clarify the point being made.
· In sentence 49, "observation" should be followed by "in" instead of "in many cases."
· In sentence 51, "ideal study" should be hyphenated.
· In sentence 56, "convection develop- ment" should be "convection development".
Reply:
Reviewer comment (R.C.): The Overview of the April 18-19 squall line case section contains several grammatical errors:
· In line 67, "It is" can be changed to "This is" for better clarity.
· In line 71, "with heavy precipitation and strong gust wind" should be changed to "with heavy precipitation and strong gusts of wind" for consistency in pluralization.
· In line 76, "occurred" can be changed to "were present" to improve clarity and avoid repetition.
· In line 78, "appeared" can be changed to "was present" to improve clarity and avoid repetition.
· In line 80, "identified" can be changed to "identified were" to improve clarity and avoid awkward phrasing.
· In line 85, "the wind" should be changed to "wind gusts" for clarity and consistency.
· In line 95, "the warm front at this level are" should be changed to "the warm front at this level is" to correct subject-verb agreement.
Reply:
Reviewer comment (R. C.): The method section could benefit from further improvement. It is important to provide a clear justification for the methodology approach used, explaining why it was chosen and how it is appropriate for the research question at hand. Additionally, it would be helpful to reference prior studies that have successfully used this methodology approach to strengthen the argument for its use in this particular study.
Reply:
Reviewer comment (R.C.): The Model configuration and experimental design section contains several grammatical errors:
· In sentence 108, add a hyphen between "Atmospheric" and "Research" to make it "Atmospheric Re-search".
· In sentence 109, add a hyphen between "horizon" and "tal" to make it "horizontal".
· In sentence 111, change "resolution" to "resolution," (with a comma) to separate it from the rest of the sentence.
· In sentence 116, add a hyphen between "squall" and "line" to make it "squall line".
· In sentence 148, add "is" after "original LH".
Reply:
Reviewer comment (R. C.): The authors need to improve the quality of the conclusions section. The conclusions section needs to be supported by the results and the authors need to show how their investigation advances the field from the present state of knowledge.
Reply:
Reviewer comment (R.C.): The Conclusion and policy recommendations section contains several grammatical errors:
· In the first sentence, "strive" should be "striving" to match the present tense used in the rest of the sentence.
· In the same sentence, "counteracting" should be "contradictory" to better convey the intended meaning.
· In the second sentence, "SDG (13)" should be "SDG 13" to match the format used in the rest of the text.
· In the third sentence, "pervaded by the informal economic activities" should be "characterized by informal economic activities" for better clarity.
· In the fourth sentence, "might prove difficult to curtail not forgetting" should be "might prove difficult to curtail, not to mention" for better grammar.
· In the seventh sentence, "the adoption cleaner energy" should be "the adoption of cleaner energy" for better grammar.
· In the eighth sentence, "while renewable energy consumption becomes an important measure for mitigating the spate of environmental pollution" should be "while the consumption of renewable energy becomes an important measure for mitigating the spate of environmental pollution" to match the structure used in the rest of the text.
· In the ninth sentence, "mollifying" should be "ameliorating" for better word choice.
· In the last sentence, "Provision of employment and infrastructural facilities in the rural area is important" should be "The provision of employment and infrastructure facilities in rural areas is important" to match subject-verb agreement.
Reply:
Reviewer comment (R. C.): The authors should consider creating a new subsection titled "Limitations and Future Recommendations". It's essential to address the study's limitations, which are the design or methodology constraints that may have affected the interpretation of the research findings. Limitations may have an impact on the ability to generalize results or describe applications for practice, as well as the usefulness of the findings that resulted from the research design or method used to establish internal and external validity, or unanticipated challenges encountered during the study.
In addition to addressing limitations, future recommendations should consider the following aspects: (1) building upon a specific finding in the research; (2) addressing a flaw in the research design; (3) testing a theory, framework, or model in a new context, location, or culture; (4) re-evaluating or (5) expanding a theory, framework, or model. It's important to consider these aspects to ensure that future research is based on solid foundations and provides valuable insights that can inform practice and policy decisions.
Reply:
Reviewer comment (R.C.): Moderate editing of English language and style required.
Reply:
Reviewer 2 Report
The study proposed in this paper is interesting, however the reasons and goals of the research are not clearly highlighted. For example, authors should briefly describe in the introductory section what are the performance benefits of the proposed cloud-resolving mmethod and how it solves the critical points of the squall lines simulation models proposed in the recent literature.
A description of the ARW-WRF model used for the simulations is needed, together with an explanation of the choice of this model.
Authors have to explain why they applied as Gaussian form the Gaussian term in equation (2) instead of using a spatial autocorrelation perturbation as in [15]. What are the performance benefits of this choice?
Figures 5 and 6 showing the thematic maps of radial reflectivity are difficult to read. I suggest separating the 8 thematic maps into two figures in order to show them four at a time with a larger size and higher resolution.
The caption of Fig. 15 is too long; I suggest shortening the description in the caption by inserting a discussion of the conceptual model in Fig. 15 in the text.
Section 6 should be completed with a description of future research perspectives.
Reviewer 3 Report
Please see the file attached.

Round 2
Reviewer 1 Report
(R.C): Accept in the present form.
Reply:
Reviewer 2 Report
The authors have taken into account all my suggestions, improving the quality of their manuscript. I consider this paper publishable in the current form.
Reviewer 3 Report
The Authors addressed all my concerns satisfactorily adding new suitable comments in the revised manuscript. I really appreciate the care with which the paper has been improved. Therefore, I'm glad to recommend the paper can be accepted in current form. Refinements in terms of style and English language can be made at the proofreading stage.